# Artificial intelligence-enhanced handheld breast ultrasound for screening: A systematic review of diagnostic test accuracy

Arianna Bunnell [1,2], Dustin Valdez[2], Fredrik Strand[3,4], Yannik Glaser[1], Peter Sadowski [1], John A. Shepherd[2]*

1 Department of Information and Computer Sciences, University of Hawai'i at Mānoa, Honolulu, Hawai'i, United States of America, 2 University of Hawai'i Cancer Center, Honolulu, Hawai'i, United States of America, 3 Department of Oncology-Pathology, Karolinska Institutet, Stockholm, Sweden, 4 Breast Radiology Unit, Medical Diagnostics Karolinska, Karolinska University Hospital, Stockholm, Sweden

* jshepherd@cc.hawaii.edu

## Abstract

Breast cancer screening programs using mammography have led to significant mortality reduction in high-income countries. However, many low- and middle-income countries lack resources for mammographic screening. Handheld breast ultrasound (BUS) is a low-cost alternative but requires substantial training. Artificial intelligence (AI) enabled BUS may aid in both the detection and classification of breast cancer, enabling screening use in low-resource contexts. The purpose of this systematic review is to investigate whether AI-enhanced BUS is sufficiently accurate to serve as the primary modality in screening, particularly in resource-limited environments. This review (CRD42023493053) is reported in accordance with the PRISMA guidelines. Evidence synthesis is reported in accordance with the SWiM (Synthesis Without Meta-analysis) guidelines. PubMed and Google Scholar were searched from January 1, 2016 to December 12, 2023. Studies are grouped according to AI task and assessed for quality. Of 763 candidate studies, 314 full texts were reviewed and 34 studies are included. The AI tasks of included studies are as follows: 1 frame selection, 6 lesion detection, 11 segmentation, and 16 classification. 79% of studies were at high or unclear risk of bias. Exemplary classification and segmentation AI systems perform with 0.976 AUROC and 0.838 Dice similarity coefficient. There has been encouraging development of AI for BUS. However, despite studies demonstrating high performance, substantial further research is required to validate reported performance in real-world screening programs. High-quality model validation on geographically external, screening datasets will be key to realizing the potential for AI-enhanced BUS in increasing screening access in resource-limited environments.

**Data availability statement:** All data are in the manuscript and/or Supporting Information files.

**Funding:** This work was supported by the National Cancer Institute (5R01CA263491 to JAS). The funders had no role in study design, data collection and analysis, decision to publish, or preparation of the manuscript.

**Competing interests:** The authors have declared that no competing interests exist.

## Author summary

Many high-income countries have seen significant decreases in breast cancer mortality through implementing mammographic screening programs. However, due to the relatively high demand on resources and personnel of mammography, many low- and middle-income countries have not implemented mammography-based screening programs. Handheld breast ultrasound (BUS) is an alternative modality that requires less equipment cost and higher portability. AI-enhanced BUS may reduce the increased burden of operator training for BUS, providing automated reporting and recommendations from imaging. As the number of commercial and academic developments of AI-enhanced BUS continues to increase, there has been no comprehensive, systematic, evaluation of model equity, performance, and readiness for use in screening. We systematically review the literature to provide a comprehensive assessment of AI-enhanced BUS model development and testing, while explicitly considering the implication of our results on screening in under-resourced contexts with few or no radiologists and sonographers. Further high-quality evidence supporting the robustness of AI-enhanced BUS for screening is needed before deployment in the clinic, particularly in resource-limited scenarios.

## Introduction

Breast cancer has become the most prevalent cancer in the world with the WHO estimating 2.3 million women diagnosed in 2020 [1,2]. High-income countries have implemented population-wide screening programs using mammography and witnessed an estimated 20% reduction in mortality in women invited for screening since the 1980s [3]. Further, regular screening with mammography is widely recommended by professional societies [4–8]. However, implementing mammographic screening is resource-intensive. Thus, many low- and middle-income countries have not been able to implement population-wide mammographic screening programs. Handheld breast ultrasound (BUS) is an alternative to mammography that requires less equipment cost and support infrastructure. Preliminary evidence from BUS screening programs in LMICs have demonstrated promising sensitivity [9,10]. However, cancer screening with BUS been found to have substantially higher false-positive rates; one representative study found a rate of 74/1,000 biopsies per screening exam with BUS alone compared to 8/1,000 with mammography alone [11]. AI-enhanced BUS may reduce the false-positive and unnecessary biopsy rate. BUS is a highly noisy, complex imaging modality which requires significant training for both image interpretation and performing exams. Importantly, AI-enhanced BUS has the potential to alleviate the need for highly trained staff, a radiologist or sonographer, to perform the examination, increasing accessibility in low-resource medical contexts [12].

For a lesion with malignancy-suspicion to be detected, the radiologist must first notice an abnormality in the ultrasound image, a perceptual task, and then assess

the probability that this lesion may be cancer, an interpretative task. Therefore, in this systematic review, we ask two questions: **Question 1 - Perception:** How accurate are AI-enhanced BUS models for frame selection, lesion detection, and segmentation when incorporated into the screening care paradigm? **Question 2 - Interpretation:** How accurate are AI-enhanced BUS models for cancer classification when incorporated into the screening care paradigm? Questions 1 and 2 are separated due to differences in performance evaluation of task types. Question 2 is concerned only with accuracy in diagnosis of lesions as benign or malignant, while Question 1 evaluates accuracy in lesion location, either alone (perception AI) or in addition to accuracy in diagnosis (perception and interpretation AI). To answer these questions, we evaluate the current literature for potential for bias in the selected studies and attribute the literature to each task-specific question to examine performance.

## Materials and methods

The abstract and full text of this systematic review are reported in accordance with the Preferred Reporting Items for Systematic Reviews and Meta-Analysis (PRISMA) guidelines (see S5 File) [13]. A protocol for this review was registered as PROSPERO CRD42023493053. We followed much of the methods of Freeman et al.'s review of AI-enhanced mammography [14]. Data extraction templates and results can be requested from the corresponding author.

### Data source, eligibility criteria, and search strategy

**Data sources, searching, and screening.** The search was conducted on PubMed [15] and Google Scholar [16] using the Publish or Perish software (Harzing, version 8). Only papers published since 2016 in English were considered and our search was updated on December 12, 2023. The search encompassed three themes: breast cancer, AI, and ultrasound. Exact search strings can be found in S1 File. Evidence on systematic review methodologies suggests the exclusion of non-English studies is unlikely to have affected results [17,18].

**Inclusion and exclusion criteria.** We included studies which reported on the performance of AI for the detection, diagnosis, or localization of breast cancer from BUS, on an unseen group of patients. Studies must additionally validate on exams from all task-relevant BI-RADS categories (i.e., BI-RADS 2 and above for classification studies). Furthermore, included studies must report a performance metric which balances sensitivity and specificity. Lastly, studies must work *automatically* from BUS images, avoiding the use of human-defined features. However, selection of a region of interest (ROI) is acceptable. Studies are additionally excluded if they include/exclude patients based on symptom presence or risk; include procedural imaging; are designed for ancillary tasks (i.e., NAC response); or are opinion pieces, reviews, or meta-analyses.

### Data collection and analysis

**Data extraction.** A single reviewer (A.B.) extracted data, subject to review by a second reviewer (D.V.) with differences resolved through discussion. The following characteristics were extracted from included articles: author(s); journal and year of publication; country of study; reference standard definition; index test definition; characteristics and count of images/videos/patients; inclusion/exclusion criteria; reader study details (if applicable); AI model source (commercial or academic); and AI and/or reader performance.

**Data synthesis.** Data synthesis is reported in accordance with the Synthesis Without Meta-analysis (SWiM) reporting guideline (see S4 File) [19]. The synthesis groupings were informed by the clinical paradigm. No meta-analysis was planned for this study as the AI tasks are heterogeneous and not well-suited for intercomparison. We utilize descriptive statistics, tables, and narrative methods. Certainty of evidence is evaluated using the following: number of studies, data split quality (if applicable), and data diversity. Heterogeneity of studies is assessed through comparison of reference standard definitions and dataset characteristics.

Studies were grouped for synthesis by clinical application time, AI task, and AI aid type (perception or interpretation). The clinical application time groups were exam time (AI is applied during BUS examination), processing time (exam recording), and reading time (pre-selected exam frames). The AI task groups and types were frame selection (perception), lesion detection (perception and interpretation), cancer classification (interpretation), and lesion segmentation (perception). In brief for this review, lesion segmentation is the pixel-wise delineation of the border of a breast lesion in a BUS exam frame which is known to have a lesion. Lesion detection is the localization of a lesion (surrounding the lesion with a bounding box) in a BUS exam frame which is not known to contain a lesion a priori. Frame selection is the filtering of BUS exam frames to those which are most informative or most likely to contain a lesion. Cancer classification is the prediction of whether a given BUS exam frame or lesion is malignant. We can define sub-groups based on the intersections of application time and task. For example, lesion detection AI applied during exam and processing time can be referred to as real-time and offline detection AI, respectively.

The outcome of interest for this review is AI performance. Lesion detection AI is evaluated by average precision (AP) or mean average precision (mAP). Both mAP and AP represent the area under the precision-recall curve, quantifying identification of true lesions balanced with prediction of false positive lesions. Frame selection is evaluated by AUROC in frame selection and/or diagnosis from selected frames. Cancer classification is evaluated by AUROC or sensitivity/specificity. AUROC is a rank-based metric which conveys the probability that a randomly selected malignant lesion will have a higher *predicted* probability of cancer than any random benign lesion. Lesion segmentation is evaluated by Dice Similarity Coefficient (DSC) or intersection over union (IOU). DSC is equal to 2 × the total *overlapping* area of two lesion segmentations, divided by the total *combined* area of the lesion segmentations. IOU is defined as the total *overlapping* area of two lesion segmentations divided by the area of their *union*. No metric conversions were attempted.

**Study quality.** Study quality was independently assessed by two reviewers (A.B. & D.V.) using the quality assessment of diagnostic accuracy studies-2 (QUADAS-2) tool [20] (see S3 File) using criteria adapted from [14]. The reviewers resolved differences through discussion. Bias criteria are rated yes, no, unclear, or not applicable. Applicability criteria are rated high, low, or unclear. Studies are classified according to their majority category. If categories are tied, the study is rated as the highest of the tied categories.

Additionally, studies are evaluated based on completeness of reporting on the racial/ethnic, age, breast density, background echotexture, and body mass index (BMI) diversity of their participants, as well as BUS machine types. Age-adjusted breast density, race/ethnicity, and BMI are known risk factors for breast cancer [21–24]. BUS machine model reporting is examined to evaluate AI generalizability.

### Changes from protocol

The addition of AUROC in diagnosis as an evaluation metric for frame selection AI was done in response to the observation that frames identified for human examination may not be most useful for downstream AI. AUROC and sensitivity/specificity were added as acceptable evaluation metrics for lesion detection AI in response to the literature. Data cleaning method was not extracted, as it was not well-defined for validation studies. Analysis by AI type was not planned but was added to emphasize clinical utility.

## Results

### Study selection

PubMed and Google Scholar yielded 322 and 709 studies, respectively. After removing duplicates, 763 articles were screened. After title (n = 242) and abstract (n = 207) exclusions, 314 full texts were evaluated. 34 studies are included. See Fig 1. screening process.

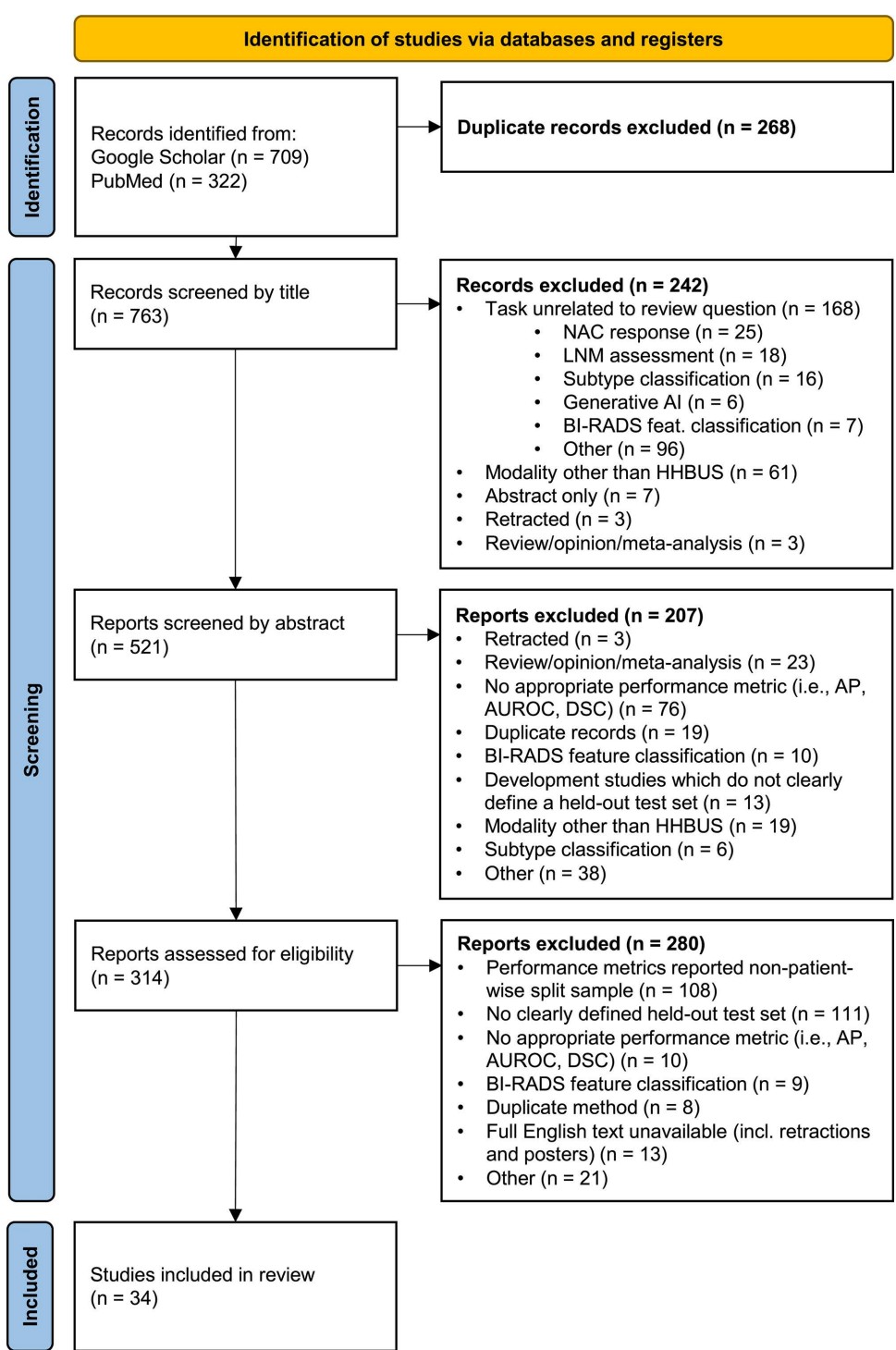

**Fig 1. PRISMA selection diagram.** PRISMA 2020 flow diagram showing study selection and screening process from PubMed and Google Scholar for perception (frame selection, lesion detection, and segmentation) and interpretation (cancer classification) breast ultrasound (BUS) AI. HHBUS = hand-held breast ultrasound; NAC = neoadjuvant chemotherapy; LNM = lymph node metastasis.

## Characteristics of included studies

The 34 included studies examined 30 AI models: 3 commercial (21% of studies), 25 academic (74%), and 2 later commercialized (6%). [25] preceded S-Detect for Breast (Samsung Medison Co., Seongnam, Korea) and [26] preceded CADAI-B (BeamWorks Inc., Daegu, Korea). Included studies analyzed a total of 5.7 million BUS images and 3,566 videos from over 185,000 patients. 5.44 million (95%) images and 143,203 patients are contributed by a single article [27]. A majority (59%) of studies were conducted in the East Asia region (20 studies; 12 in China). 5 studies used only public datasets (see S2 File).

## AI Tasks

There were 6 lesion detection studies [26,28–32], 1 frame selection study [33], 16 classification studies (12 AI models) [25,27,34–47], and 11 segmentation studies [48–58]. 18 studies use *perception* AI [26,28–33,48–58] and 22 studies use *interpretation* AI [25–32,34–47], with 6 studies [26,28–32] using AI for both.

**Perception: Frame selection (1 study).** Frame selection AI models identify exam frames for downstream examination of lesion location and cancer status. See Table 1 (bottom) for a summary. Huang 2022a develop a reinforcement learning model, rewarded by optimizing identifying frames likely to contain lesions, annotations, and malignancies. Their model increased diagnostic performance of senior and junior readers by 0.03 and 0.01 AUROC, respectively.

**Perception: Lesion segmentation (11 studies).** Lesion segmentation AI models delineate lesions for downstream evaluation of cancer status. See Table 1 for a summary. Six (55%) and nine (82%) studies train and test on at least partially public data. The most common approach was extending the U-Net [59] architecture (seven studies, 64%). Reported DSC ranges from 0.701 [48] to 0.872 [52] on test datasets ranging from 42 [49] to 1,910 [57] images. The remaining studies develop convolutional [53,55], graph convolutional [51], and adversarial networks [50]. Han 2020 report 0.78 DSC on an external test dataset. Huang 2022b and Qu 2020 report 0.919 and 0.905 DSC on five-fold cross-validation. Webb 2021 report 0.832 DSC on an internal test set of 121 images (85 patients).

**Interpretation: Cancer classification (16 studies).** Cancer classification AI models classify lesions/images as either benign or cancerous. See Table 2 for a summary. Operator involvement required prior to AI use varied: six studies (38%) require ROI selection, three studies require seed point placement (19%), three studies (19%) require image hand-cropping, three studies (19%) apply automatic cropping/segmentation, and one study (6%) is unclear. Choi 2019, Lee 2022, and Park 2019 test S-Detect for Breast (Samsung Medison Co., Seongnam, Korea). Choi 2019 and Lee 2022 find standalone AI to perform with 85% and 86.2% sensitivity and 95.4% and 85.1% specificity, respectively. Park 2019 find AI assistance to increase reader sensitivity by 10.7% and specificity by 8.2%. Han 2017 finetune GoogLeNet [60] and report 0.96 AUROC on an internal dataset. Berg 2021, Guldogan, and Wanderley 2023 all validate Koios DS (Koios Medical, Inc., Chicago IL) through reader studies. Berg 2021 find standalone AI performs with 0.77 AUROC. Guldogan 2023 and Wanderley 2023 evaluate binned predictions and find AI alone performs with 98.5% and 98.2% sensitivity and 65.4% and 39% specificity, respectively. The nine remaining studies develop AI models. Reported AUROC values range from 0.81 [41] to 0.98 [27] on test datasets ranging from 33 [40] to 25,000 [27] patients. The most common approach was to finetune and optionally extend an existing architecture from ImageNet [61] weights. Otherwise, studies used generative adversarial networks [37] and custom convolutional architectures [27]. All studies except Liao 2023 explicitly work on unenhanced (B-mode) BUS images. Fig 2 displays reported performance vs. development dataset size. Only two studies developed on datasets with over 20,000 images, performing with 0.91 [47] and 0.976 [27] AUROC.

**Perception and interpretation: Lesion detection (6 studies).** Lesion detection AI models perform both detection and cancer classification of lesions. See Table 3 for a summary. Lesion localization precision varied: a single study provides heatmap-style visualizations [26], three studies provide bounding boxes [29–31], and two studies provide delineations [28,32]. Qiu 2023, Meng 2023, and Fujioka 2023 all extend the YOLO family [62] and achieve 0.87 AUROC (no location performance measure) on 278 videos, 0.78 mAP on 647 images, and an increase in per-case sensitivity and specificity of

**Table 1. Lesion segmentation and frame selection study details.**

**Lesion segmentation**

| Study | Population | Reference Standard | Index Test | Performance |
|---|---|---|---|---|
| Byra 2020 | 882 images of 882 lesions (? malignancy) from? patients from a single clinical site. **External Testing:** BUSI UDIAT and OASBUD. | Delineations from a single "medical expert." | Adapted U-Net with additionally dilated convolution blocks replacing traditional convolution blocks. | 0.701 mean DSC when not finetuned on external test sets. |
| Chen 2023 | BUSI and UDIAT (42% malignancy) **External Testing:** STU-Hospital (? malignancy). | | Adapted U-Net with attention modules with varying sizes replacing traditional convolution blocks. | 0.802 DSC on external test set. |
| Han 2020 | 2,800 images from 2,800 patients (50% malignancy) from a single hospital in China. **External Testing:** UDIAT | Delineations from physicians at department of US. | GAN-based architecture with attention and segmentation mask generator and discriminator networks. | 0.78 DSC on external test set |
| Huang 2022b | 2,020 images from? patients (50.2% malignancy) from UDIAT and a single hospital in China. | Delineations from "experienced radiologist." | Combination CNN and graph convolutional architecture for mask and specific boundary-rendering, respectively. | 0.919 DSC on 5-fold CV. |
| Ning 2022 | UDIAT, BUSI, and ultrasoundcases. **External Testing:** onlinemedicalimages and OASBUD. | | Custom U-Net with background/foreground information streams and shape-, edge-, and position-aware fusion units. | 0.872 DSC on external test set. |
| Qu 2020 | 980 images from 980 women (60.7% malignancy) from a single university hospital in China and UDIAT. | Delineations from "experts." | Custom ResNet with varying-scale attention modules and upsampling. | 0.905 DSC on five-fold CV. |
| Wang 2021 | 3,279 images from 1,154 patients (57.2% malignancy) (ultrasoundcases & BUSI). **External Testing:** BUSI & radiopaedia. | | Custom U-Net with ResNet34 encoder and residual feedback. | 0.82 DSC on external test set. |
| Webb 2021 | 31,070 images from 851 women (? malignancy) from a single clinic in the USA | Delineations from 3 "experts" (testing) and "research technologists" with US experience (development). | Custom DenseNet264 with added feature pyramid network and ResNet-C input stream pretrained on thyroid US images. | 0.832 DSC on internal test set. |
| Zhang 2023 | 1,342 images from? patients from 5 hospitals in China. **External Testing:** 570 images from? patients from a single hospital in China & BUSI & onlinemedicalimages. | Delineations from "experienced radiologists." | Combination U-Net and DenseNet backbone from pre-selected ROI. | 0.89 mean IOU on external test set |
| Zhao 2022 | 9,836 images from 4,875 patients from? hospitals in China. | Delineations are from 3 "experienced radiologists." | Custom U-Net architecture with local and de-noising attention. | 0.838 DSC on internal test set |
| Zhuang 2019 | 857 images from? patients from a single hospital in the Netherlands (ultrasoundcases). **External Testing:** STU-Hospital & UDIAT. | | Custom attention-based residual U-Net. | 0.834 DSC on external test set |

**Frame Selection**

| Study | Population | Reference Standard | Index Test | Performance |
|---|---|---|---|---|
| Huang 2022a | 2,606 videos from 653 patients (26.7% malignancy) from 8 hospitals in China | **Keyframe/Location:** Frame and bounding box from "experienced sonographers" **Classification:** Histological results from biopsy or surgery | Reinforcement learning scheme with 3D convolutional BiLSTM with frame-based reward structure based on lesion presence, proximity to labelled frame, and malignancy indicators. | 0.793 diagnostic AUROC on internal test set from selected frames |

S2 File provides a complete accounting of public datasets (i.e., UDIAT, BUSI, OASBUD). CV = cross-validation; DSC = Dice similarity coefficient; CNN = convolutional neural network; GAN = generative adversarial network; US = ultrasound. Unknown values (not reported in study) are indicated with a "?" symbol.

**Table 2.** Lesion classification study details.

| Study | Population | Reference Standard | Index Test | Performance |
|---|---|---|---|---|
| Berg 2021 | **External Testing:** 638 images of 319 lesions (27.5% malignancy) from? women from a single health center in the US | Histological results from biopsy with benign follow-up of at least 2 years | Koios DS from pre-selected ROI | 0.77 AUROC of AI alone on external test set |
| Byra 2019 | 882 images from 882 patients (23.1% malignancy) from a single health center in California. **External Testing:** UDIAT & OASBUD | Histological results from biopsy with benign follow-up of at least 2 years | SVM from finetuned VGG19 pretrained on ImageNet from pre-selected ROI | 0.893 AUROC on external test set |
| Choi 2019 | **External Testing:** 759 images of 253 lesions from 226 patients (31.6% malignancy) from a single medical center in South Korea | Histological results from biopsy with benign follow-up of? | S-Detect for Breast | 85.0% sensitivity and 95.4% specificity for AI alone |
| Fujioka 2020 | 702 images from 217 patients (48.9% malignancy in testing) from a single health center in Japan | Histological results from biopsy with benign follow-up of at least 1 year | Bidirectional GAN from hand-cropped images | 0.863 AUROC on internal test set |
| Gu 2022 | 11,478 images from 4,149 patients (42.7% malignancy) from 30 tertiary-care hospitals in China **External Testing:** 1,291 images from 397 patients (62.1% malignancy) from 2 tertiary-care hospitals in China & BUSI | Histological results from biopsy or surgery | Finetuned VGG19 backbone pretrained on ImageNet from pre-selected ROI | 0.913 AUROC on external test set |
| Guldogan 2023 | **External Testing:** 1,430 orthogonal images of 715 lesions (18.8% malignancy) from 530 women | Histological results from biopsy with benign follow-up of at least 2 years | Koios DS from pre-selected ROI | 98.5% sensitivity and 65.4% specificity for AI alone |
| Han 2017 | 7,408 images from 5,151 patients (42.6% malignancy) from a single health center in South Korea | Histological results from biopsy | Finetuned GoogLeNet pretrained on grayscale ImageNet from semi-automatic segmentation | 0.958 AUROC on internal test set |
| Hassanien 2022 | UDIAT | | Finetuned SwinTransformer from hand-cropped images | 0.93 AUROC on internal test set |
| Karlsson 2022 | BUSI **External Testing:** 293 images from? women (90.1% malignancy) from a single university hospital in Sweden | | Finetuned ResNet50V2 from hand- and automatically-cropped images | 0.81 AUROC on external test set |
| Lee 2022 | **External Testing:** 492 lesions from 472 women (40.7% malignancy) from a single health center in South Korea | Histological results from biopsy with benign follow-up of at least 2 years | S-Detect for Breast | 0.855 AUROC on external test set |
| Liao 2023 | 15,910 images from 6,795 patients (2.56% malignancy) from a single hospital in China **External Testing 1:** 896 images from 391 patients (2.23% malignancy) from a single hospital in China **External Testing 2:** 490 images from 235 patients (2.04% malignancy) from a single hospital in China | Histological results from biopsy with benign follow-up of at least 3 years | 80 Dual-branch ResNet50 learners for B-mode and Doppler ensembled into parent model | 0.956 AUROC on external test set |
| Park 2019 | **External Testing:** 100 video clips of lesions from 91 women (41% malignant) from a single hospital in South Korea | Histological results from biopsy or surgery | S-Detect for Breast | +0.105 difference in AUROC with/without AI for readers on external test set |
| Shen 2021 | 5,442,907 images from 143,203 patients (1.1% malignancy) from >100 hospitals in New York **External Testing**: BUSI | Histological results from biopsy with benign follow-up of at most 15 months (test set); Pathology report (training set) | Deep convolutional network with spatial and scan-wise attention and saliency map concatenation from entire input image set per breast | 0.976 AUROC on internal test set |

*(Continued)*

**Table 2.** (Continued)

| Study | Population | Reference Standard | Index Test | Performance |
|---|---|---|---|---|
| Wanderley 2023 | **External Testing:** 555 lesions from 509 women (40% malignancy) from a single health center in Brazil | Histological results from biopsy | Koios DS from pre-selected ROI | 98.2% sensitivity and 39.0% specificity of CAD alone on external test set |
| Wu 2022 | 13,684 images from 3,447 patients (28.7% malignancy) from a single hospital in China **External Testing:** 440 images from 228 patients (54.3% malignancy) from a single hospital in China | Histological results from biopsy or surgery | Finetuned MobileNet from hand-cropped images. | 0.893 AUROC on external test set |
| Xiang 2023 | 39,899 images of 8,051 lesions from 7,218 patients (64.1% malignancy) from a single university hospital in China **External Testing 1:** 2,637 images of 777 lesions from 693 patients (47.6% malignancy) from a single hospital in China **External Testing 2:** 957 images of 419 lesions from 382 patients (48.9% malignancy) from a single hospital in China **External Testing 3:** 2,416 images of 648 lesions from 504 patients (25.3% malignancy) from a single hospital in China | Histological results from biopsy or surgery | Custom finetuned DenseNet121 with self-attention averaged over all views of a lesion | 0.91 AUROC on external test set |

S2 File provides a complete accounting of public datasets (i.e., UDIAT, BUSI, OASBUD). AUROC = area under the receiver operating characteristic curve; ROI = region of interest; CAD = computer-aided diagnosis. Unknown values (not reported in study) are indicated with a "?" symbol.

11.7% and 20.9% (reader study) on 230 videos, respectively. Kim 2021b extend the GoogLeNet [60] architecture to achieve 0.9 AUROC and 99% correct localization on an external dataset of 200 images. Lai 2022 evaluate standalone BU-CAD (TaiHao Medical Inc., Taipei City, Taiwan) on 344 images, resulting in a location-adjusted AUROC of 0.84. Bunnell 2023 develop an extension to the Mask RCNN [63] architecture and achieve mAP 0.39 on an internal test dataset of 447 images.

## Clinical application time

We define an example care paradigm inclusive of low-resource, teleradiology-exclusive medical scenarios. See Fig 3. The clinical application time of studies included 5 exam, 2 processing, and 27 reading time studies.

## Study quality assessment

Fig 4 displays bias assessment results. 18 (53%) and 9 (27%) studies have high or unclear risk of bias overall. All studies but one are of high applicability concern. Concerns about applicability for Qiu 2023 are attributed to an unclear location reference standard. Generally, studies are at an unclear risk of bias and high applicability concern for patient selection due to incomplete reporting of the participant selection process. All included studies except Liao 2023 and Shen 2021 are of high index test applicability concern due to making image-level predictions only. Studies which aggregate predictions into exam-, breast-, or patient-level predictions have lower index test applicability concern. Risk of bias in participant selection was also high due to unrepresentative dataset composition; only two studies (Liao 2023 and Shen 2021) trained or validated their AI methods on datasets with screening cancer prevalence (<3%).

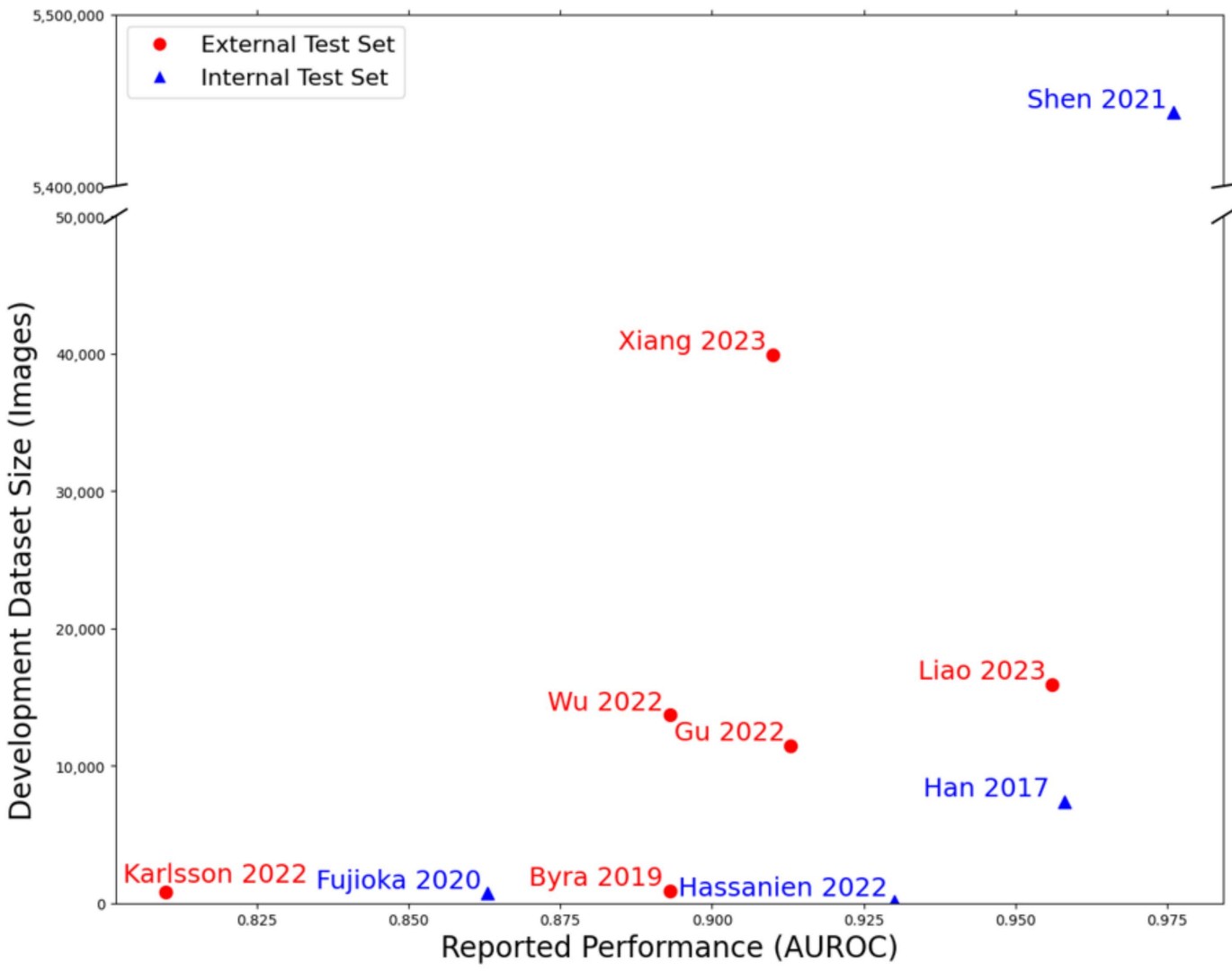

**Fig 2. Performance of lesion classification studies.** Scatter plot showing reported performance (as measured by AUROC) for lesion classification (interpretation) studies against the reported size of the development dataset by number of breast ultrasound images. Studies are additionally identified by whether reported performance is on an internal or external testing set. Internal testing sets are sampled from the same underlying population as the development set.

See Fig 5 for a complete breakdown of diversity reporting. 35% of included studies failed to report diversity along any axis. The most reported diversity axes were participant age (15 studies) and machine type (18 studies). Classification studies were the most complete, with 11 (69%) reporting along at least one axis.

## Discussion

### Main findings

In this systematic review, we evaluated the accuracy of BUS AI models for each identified task. We identified 6 studies performing lesion detection, 1 frame selection study, 16 cancer classification studies, and 11 lesion segmentation studies. 12 studies aid in perceptual tasks, 16 studies aid in interpretative tasks, and 6 studies aid in both. We also examine

**Table 3. Lesion detection study details.**

| Study | Population | Reference Standard | Index Test | Performance |
|---|---|---|---|---|
| Bunnell 2023 | 37,921 images from 2,148 women (24.2% malignancy) from? clinical sites in the US. | **Location:** Delineations from a single radiologist. **Classification:** Histological results from biopsy with no record of cancer for benign. | Finetuned Mask-RCNN with ResNet-101 backbone and custom heads for BI-RADS mass feature prediction. | 0.39 mAP on internal test set. |
| Fujioka 2023 | 88 videos from 45 women (? malignancy) from a single breast surgery department in Japan. **Internal Testing:** 232 videos (40.5% malignancy) from 232 women from a single breast surgery department in Japan. | **Location:** 2 experts (>10 years of BUS experience). A 3rd expert then performed adjudication. **Classification:** Unclear. | Finetuned YOLOv3-tiny combined with edge detection post-processing of regions to isolate lesions. | 95.5% sensitivity and 2.2% specificity for AI alone. |
| Kim 2021b | 1,400 images from 971 patients (50% malignancy) from a single university hospital in South Korea. **External Testing:** 200 images from 125 patients (50% malignancy) from a single university hospital in South Korea. | **Location:** Delineations from a single radiologist. **Classification:** Histological results from biopsy with benign follow-up of at least 2 years. | GoogLeNet from hand-cropped images with saliency maps for localization. | 0.9 AUROC on external test set. |
| Lai 2022 | **External Testing:** 344 images from 172 women (37.8% malignancy) from a single hospital in Taiwan. | **Location:** From "expert panel" of 5 radiologists. **Classification:** Histological results from biopsy with benign follow-up of at least 2 years. | BU-CAD (TaiHao Medical Inc., Taipei Taiwan) | 0.838 AULROC on external test set. |
| Meng 2023 | 7,040 images from 3,759 women (60.7% malignancy) from? hospitals in China. **External Testing:** BUSI | **Location:** Delineations from "experienced radiologists." **Classification:** Histological results from biopsy. | Adapted YOLOv3 with added bilateral spatial and global channel attention modules. | 0.782 mAP on external test set. |
| Qiu 2023 | 480 video clips (18,122 images) of 480 lesions from 420 women (40.8% malignancy) from a single hospital in China **Prospective Testing:** 292 video clips of 292 lesions from 278 women (42.5% malignancy) from 2 hospitals in China | **Location:** Delineations from 2 "experienced radiologists." **Classification:** Histological results from biopsy. | Finetuned YOLOv5 network with attention | 0.87 AUROC on prospective testing set |

S2 File provides a complete accounting of public datasets (i.e., UDIAT, BUSI, OASBUD). *mAP = mean average precision; AUROC = area under the receiver operating characteristic curve.* Unknown values (not reported in study) are indicated with a "?" symbol.

clinical application time in the screening care paradigm: 5 studies were designed for exam time, 2 for processing time, and 27 for reading time. Included studies examined the following commercial systems, as well as 25 academic models: S-Detect for Breast (Samsung Medison Co., Seongnam, Korea) (4 studies), CADAI-B (BeamWorks Inc., Daegu, Korea) (1 study), BU-CAD (TaiHao Medical Inc., Taipei City, Taiwan) (1 study), and Koios DS (Koios Medical, Inc., Chicago IL) (3 studies). Koios DS is the only system included in this review with current US FDA clearance. Overall, the current state-of-the-art in AI-enhanced BUS for frame selection, lesion detection, and lesion segmentation (perception) does not yet provide evidence that it performs sufficiently well for integration into breast cancer screening where BUS is the primarily modality, particularly when not supervised at all stages by a radiologist (Question 1). Zhao 2022 provide the highest-quality perceptual evidence, reporting 0.838 DSC on an internal test dataset of 1,910 images. The included studies report high performance but lack sufficient validation and population reporting and commonly validate on datasets unrepresentative of screening (<3% cancer prevalence). Models trained on datasets enriched with malignancies require an additional calibration step before use in the screening population. Validation of models on larger datasets containing more normal/benign imaging, as well as unaltered BUS video, would improve evidence supporting these models.

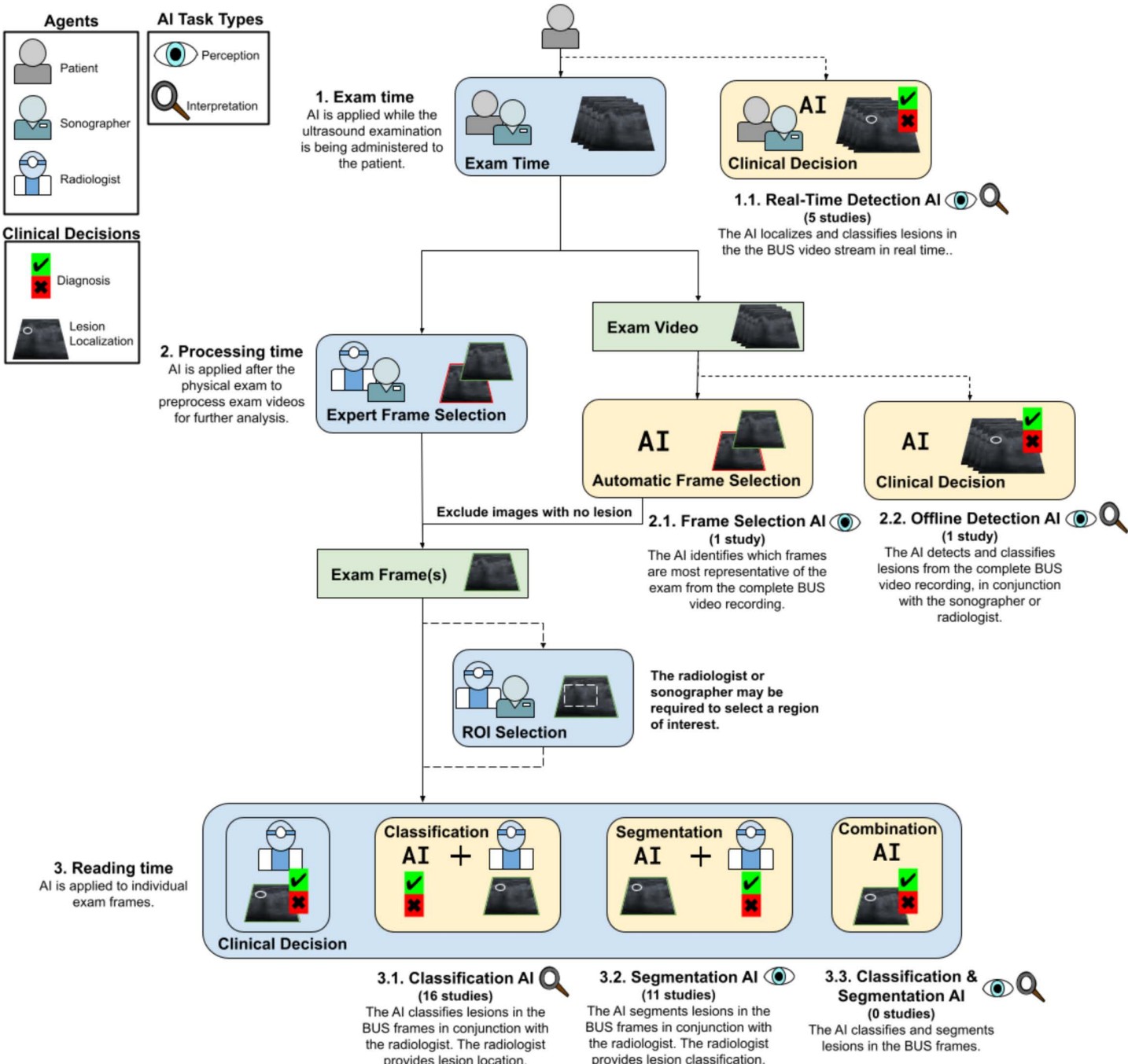

**Fig 3. Process diagram showing clinical application time and AI task types of studies.** Diagram showing the different opportunities in the care paradigm where AI can be applied in a low-resource or teleradiology-first setting. Agents include the patient and healthcare team (sonographer and/ or radiologist). Clinical decisions are lesion diagnosis or localization. Studies are classified as being interpretation (cancer classification) or perception (frame selection, lesion detection, and segmentation) according to their AI task type (one of: real-time detection, offline detection, frame selection, classi- fication and/or segmentation).

| Study (Top) | Risk of Bias | | | | Applicability Concerns | | | Study (Bottom) |
|---|---|---|---|---|---|---|---|---|
| | Patient Selection | Index Test | Reference Standard | Flow and Timing | Patient Selection | Index Test | Reference Standard | |
| **AI Classification Systems** | | | | | | | | |
| Berg 2021 | Low | High | Low | High | High | High | Low | Byra 2019 |
| | Unclear | High | Low | High | High | High | Low | |
| Choi 2019 | High | Low | High | High | High | High | High | Fujioka 2020 |
| | High | Unclear | High | High | High | High | High | |
| Gu 2022a | Unclear | Low | High | High | High | High | High | Guldogan 2023 |
| | Low | Low | High | High | High | High | High | |
| Han 2017 | Unclear | Low | High | High | High | High | High | Hassanien 2022 |
| | Unclear | Low | High | High | High | High | Unclear | |
| Karlsson 2022 | Unclear | Low | High | Low | High | High | High | Lee 2022 |
| | Low | Low | Low | High | High | High | Low | |
| Liao 2023 | Low | High | Low | Low | High | High | Low | Park 2019 |
| | Unclear | Low | High | Low | High | High | High | |
| Shen 2021 | Unclear | Low | High | High | High | Low | High | Wanderley 2023 |
| | Unclear | High | High | Low | High | High | High | |
| Wu 2022 | Unclear | High | High | High | High | High | High | Xiang 2023 |
| | Unclear | High | High | Low | High | High | High | |
| **AI Segmentation Systems** | | | | | | | | |
| Byra 2020 | Unclear | High | High | Unclear | High | High | High | Chen 2023 |
| | Low | Low | High | Unclear | High | High | High | |
| Han 2020b | Unclear | Low | High | Unclear | High | High | High | Huang 2022a |
| | High | Unclear | High | Unclear | High | High | High | |
| Ning 2022 | Unclear | Low | High | Unclear | High | High | High | Qu 2020 |
| | Unclear | Low | High | Unclear | High | High | Unclear | |
| Wang 2021 | Unclear | Low | High | Unclear | High | High | High | Webb 2021 |
| | High | High | High | Unclear | High | High | High | |
| Zhang 2023 | Unclear | Low | High | Unclear | High | High | High | Zhao 2022 |
| | Unclear | Low | Low | Unclear | High | High | Low | |
| Zhuang 2019 | Unclear | Low | High | Unclear | High | High | High | |
| **Real-Time Detection AI** | | | | | | | | |
| Bunnell 2023 | High | Low | High | High | High | High | High | Fujioka 2023 |
| | Unclear | High | High | High | High | Low | High | |
| Kim 2021 | Unclear | Low | High | High | High | High | High | Lai 2022 |
| | Unclear | Low | Unclear | High | High | High | High | |
| Meng 2023 | Unclear | Low | High | Low | High | High | High | |
| **Offline Detection AI** | | | | | | | | |
| Qiu 2023 | Unclear | Low | High | Low | High | Low | Unclear | |
| **Frame Selection AI** | | | | | | | | |
| Huang 2022b | Unclear | Low | H / U | Unclear | High | Low | H / U | |

**Fig 4. QUADAS-2 bias assessment results.** QUality Assessment of Diagnostic Accuracy Studies-2 (QUADAS-2) bias assessment results. Figure is best viewed in color. Studies are assessed for risk of bias in patient selection, index test (i.e., AI model) definition, reference standard (ground truth) definition, and the flow and timing of their study in relation to decisions about patient care. Studies are assessed for concerns about applicability for patient selection, index test definition, and reference standard definition. Reference standard assessments for frame selection studies are reported classification first, frame selection second. H = high; U = unclear. The full QUADAS-2 criteria adapted from [14] is available in the S3 File.

Many more high-quality studies develop cancer classification AI, forming a more robust picture of interpretation AI performance (Question 2). We refer to Shen 2021, Xiang 2023, and Liao 2023 as the best examples, showing performances of 0.976, 0.91, and 0.956 AUROC (respectively) on large datasets. We suggest that validation of BUS cancer classification AI on a common dataset with comprehensive patient metadata and containing more normal/benign imaging may facilitate easier comparison between methods, allowing for a more complete picture of the state of the field on subgroups of interest.

We find that 79% of included studies are at high or unclear risk of bias. The main sources of bias observed were: (1) unclear source of ground truth for lesion location; (2) incomplete reporting of the patient/image selection process; and (3) failure to aggregate image-level results into exam- or woman-level predictions. Furthermore, the lack of external model validation, on imaging from new populations of women at different institutions, is a key weakness of the current literature in AI-enhanced BUS. Prospective validation on a racial/ethnically diverse, external population of women represents the

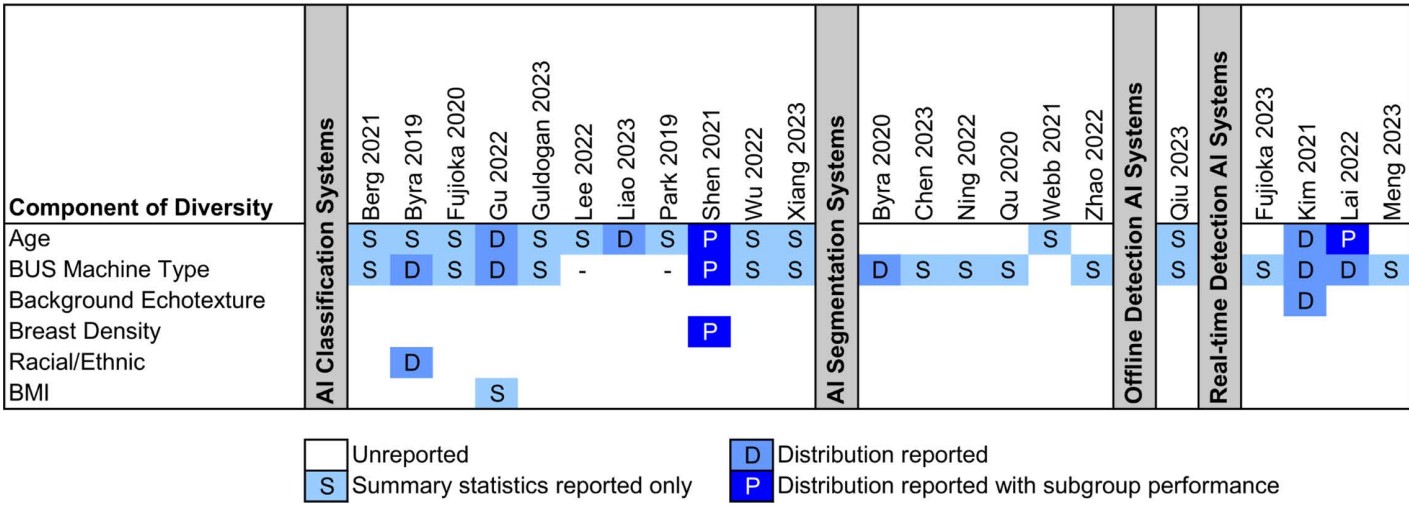

**Fig 5. Diversity reporting completeness heatmap.** Heatmap showing axes of reported diversity for included studies. Studies are evaluated based on completeness of reporting on the racial/ethnic, age, breast density, background echotexture, and body mass index diversity of their participants, as well as included breast ultrasound machine types. Figure is best viewed in color. Studies which fail to report along any of the included axes are omitted from the plot. Studies which only use one kind of ultrasound machine and report on an additional axis are indicated with a – on the above plots.

gold standard in model evaluation. None of the included studies perform this style of validation. Lack of reporting on data diversity is also a concern, limiting evidence for model generalizability. While the majority of studies report patient age and BUS machine types, very few studies report patient BMI, racial/ethnic distribution, breast density, and background echotexture (Byra 2019, Gu 2022, Shen 2021, and Kim 2021).

**Comparison with other studies.** Although others have reviewed AI-enhanced BUS [64–76], we contribute the first *systematic* review not limited to a single BUS modality, as in [70], and contribute the only QUADAS-2 bias assessment of AI for BUS. [14] serves as a close analog to this work, examining test accuracy in mammography AI. However, [14] excludes all studies which evaluate performance on split sample datasets. This strict validation criteria improves the evidence supporting model performance in new patient populations and represents the highest level of dataset split quality. We remove this restriction due to the relatively early stage of the field of BUS AI development as compared to mammography AI. For example, the FDA approved the first mammography CAD system in 1998 [77], whereas the first BUS CAD system wasn't approved until 2016 [78]. In initial stages, more AI models may be developed and validated within a single institution.

### Strengths and limitations

We followed conventional methodology for systematic reviews and applied strict inclusion criteria to ensure the reliability and quality of the included studies. Studies using internal validation on the image-, video-, or lesion-level, or no held-out testing set do not provide good evidence of model generalizability. By upholding strict standards for model validation, we attempt to provide a clear picture of AI performance. However, we did not apply exclusion criteria based on dataset size, thus our review is limited in inclusion of studies with small testing sets, which provide poor evidence of generalizability. Lastly, we are limited in that we consider the application of QUADAS-2 guidelines in the manner of [14], but do not evaluate with a bias framework specific for medical AI studies, such as QUADAS-2 for AI [79] or STARD-AI [80], both of which are yet to be published. CONSORT-AI [81] and DECIDE-AI [82] were not applicable as included studies are not clinical trials or evaluated online. This review is limited in that there may be unidentified AI tasks which exist within the screening paradigm. One example of this may be AI designed to verify coverage of the entire breast during BUS scanning.

We conclude that high accuracy can be obtained in both perception and interpretation BUS AI. However, researchers developing AI-enhanced BUS systems should concentrate their efforts on providing explicit, high-quality model validation on geographically external test sets, with breast cancer prevalence representative of screening, with complete metadata. Creation of a secure benchmark dataset which meets these criteria may is one promising method by which new models can be evaluated, and this would be helpful in advancing the field. Studies should emphasize the entire clinical workflow. For example, real-time detection methods for low-resource settings must have performance reported on a dataset of complete BUS exam frames from a geographically external set of participants, imaged by non-experts, rather than on curated or randomly-selected frames. Considering the potential for AI-enhanced BUS to improve access to breast cancer screening in low- and middle-income countries in particular, the absence of a radiologist or experienced breast sonographer to additionally examine all imaging limits the safeguards we can assume are in place in the clinic, adding to the urgency of more complete, high-quality performance and metadata reporting for BUS AI across the clinical paradigm.

## Supporting information

**S1 File. Google Scholar and PubMed complete search strings.** Complete search strings for PubMed and Google Scholar searches.
(PDF)

**S2 File. Public BUS datasets.** Complete list and description of data characteristics of public datasets referenced by name in the main text.
(PDF)

**S3 File. Complete QUADAS-2 criteria.** QUality Assessment of Diagnostic Accuracy Studies-2 (QUADAS-2) criteria used to assess the risk of bias for this systematic review, adapted from [14].
(PDF)

**S4 File. SWiM reporting sheet.** Completed Synthesis Without Meta-analysis (SWiM) [19] reporting guideline.
(PDF)

**S5 File. PRISMA reporting sheet.** Completed Preferred Reporting Items for Systematic Reviews and Meta-Analysis (PRISMA) [13] guidelines.
(PDF)

**S6 File. List of exclusions.** Complete list of studies excluded after full-text review with reasons for exclusion (after adjudication).
(PDF)

**S7 File. Data Extraction Sheet.** Data extraction results for included studies.
(PDF)

**S8 File. Complete QUADAS-2 results.** For each included study, results from the QUality Assessment of Diagnostic Accuracy Studies-2 (QUADAS-2) criteria used for this review.
(PDF)

## Author contributions

**Conceptualization:** Arianna Bunnell, Peter Sadowski, John A. Shepherd.

**Data curation:** Arianna Bunnell, Dustin Valdez.

**Formal analysis:** Arianna Bunnell.

**Funding acquisition:** John A. Shepherd.

**Investigation:** Arianna Bunnell, Fredrik Strand.

**Methodology:** Arianna Bunnell.

**Resources:** Peter Sadowski, John A. Shepherd.

**Supervision:** Peter Sadowski, John A. Shepherd.

**Validation:** Yannik Glaser.

**Visualization:** Arianna Bunnell, Yannik Glaser.

**Writing – original draft:** Arianna Bunnell.

**Writing – review & editing:** Dustin Valdez, Fredrik Strand, Yannik Glaser, Peter Sadowski, John A. Shepherd.

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
