## [Decision Letter · Decision Letter 0]

3 Jun 2025

PDIG-D-25-00126Artificial Intelligence-Informed Handheld Breast Ultrasound for Screening: A Systematic Review of Diagnostic Test AccuracyPLOS Digital Health Dear Dr. Bunnel, Thank you for submitting your manuscript to PLOS Digital Health. After careful consideration, we feel that it has merit but does not fully meet PLOS Digital Health's publication criteria as it currently stands. Therefore, we invite you to submit a revised version of the manuscript that addresses the points raised during the review process. Please submit your revised manuscript within 30 days Jul 03 2025 11:59PM. If you will need more time than this to complete your revisions, please reply to this message or contact the journal office at digitalhealth@plos.org. Please include the following items when submitting your revised manuscript: * A rebuttal letter that responds to each point raised by the editor and reviewer(s). You should upload this letter as a separate file labeled 'Response to Reviewers '. This file does not need to include responses to any formatting updates and technical items listed in the 'Journal Requirements' section below.* A marked-up copy of your manuscript that highlights changes made to the original version. You should upload this as a separate file labeled 'Revised Manuscript with Track Changes '.* An unmarked version of your revised paper without tracked changes. You should upload this as a separate file labeled 'Manuscript '. If you would like to make changes to your financial disclosure, competing interests statement, or data availability statement, please make these updates within the submission form at the time of resubmission. Guidelines for resubmitting your figure files are available below the reviewer comments at the end of this letter. We look forward to receiving your revised manuscript. Kind regards, Erika OngAcademic EditorPLOS Digital Health Shrey LakhotiaAcademic EditorPLOS Digital Health Leo Anthony CeliEditor-in-ChiefPLOS Digital Healthorcid.org/0000-0001-6712-6626

**Journal Requirements:**

1. Please provide a complete Data Availability Statement in the submission form, ensuring you include all necessary access information or a reason for why you are unable to make your data freely accessible. If your research concerns only data provided within your submission, please write "All data are in the manuscript and/or supporting information files" as your Data Availability Statement.

2. Please provide separate figure files in .tif or .eps format.

3. As required by our policy on Data Availability, please ensure your manuscript or supplementary information includes the following: 

**Additional Editor Comments (if provided):**

Please address the reviewers' comments. Thank you.

**Reviewers' Comments:**

Reviewer's Responses to Questions

**Comments to the Author**

1. Does this manuscript meet PLOS Digital Health’s publication criteria ? Is the manuscript technically sound, and do the data support the conclusions? The manuscript must describe methodologically and ethically rigorous research with conclusions that are appropriately drawn based on the data presented.

Reviewer #1: Yes

Reviewer #2: Yes

Reviewer #3: Yes

2. Has the statistical analysis been performed appropriately and rigorously?

Reviewer #1: Yes

Reviewer #2: N/A

Reviewer #3: Yes

3. Have the authors made all data underlying the findings in their manuscript fully available (please refer to the Data Availability Statement at the start of the manuscript PDF file)?

Reviewer #1: Yes

Reviewer #2: Yes

Reviewer #3: Yes

4. Is the manuscript presented in an intelligible fashion and written in standard English?

Reviewer #1: Yes

Reviewer #2: Yes

Reviewer #3: Yes

5. Review Comments to the Author

Reviewer #1: This is an important and timely systematic review addressing a key gap in the field—namely, the accuracy and readiness of AI-informed handheld breast ultrasound (BUS) for screening in resource-limited settings. The manuscript has several strengths:

Strengths:

1. The protocol registration (PROSPERO), use of PRISMA and SWiM, and quality assessment (QUADAS-2) all strengthen the credibility and transparency of the review.

2. The clear distinction between AI used for perceptual (e.g., segmentation, detection) and interpretive (e.g., classification) tasks, along with integration into clinical workflow stages (exam, processing, reading), adds valuable structure and relevance for both clinical and technical audiences.

3. The attention given to dataset diversity—including age, race, BMI, and machine type—strengthens the manuscript’s relevance for global health equity and AI fairness.

Suggestions for Improvement:

1. Consider clarifying terminology for a broader digital health audience—for example, explain “Dice Similarity Coefficient” briefly where first mentioned.

2. While limitations are acknowledged, consider emphasising how the high prevalence of malignancy in most studies (unrealistic for screening populations) may distort diagnostic performance.

3. Another potential limitation of this review is the restricted number of databases used in the literature search (PubMed and Google Scholar only). While these are important sources, the exclusion of other major databases such as Embase, Scopus, or Web of Science might have led to the omission of relevant studies, particularly from non-biomedical domains or regional sources. For future systematic reviews in this fast-evolving AI landscape, expanding the database scope is recommended to ensure a more exhaustive literature capture.

4. Enhance the self-sufficiency of figure legends so that readers can interpret the figure without needing to cross-reference the text.

5. Consider recommending more explicitly the development of a shared benchmark dataset for AI-informed BUS, which could accelerate field-wide progress.

Overall, this is a high-quality and much-needed review. With minor revisions to improve clarity and emphasis on applicability, it will be a valuable resource for researchers, clinicians, and policymakers aiming to implement AI solutions in low-resource breast cancer screening programs.

Reviewer #2: This systematic review addresses the diagnostic accuracy of AI-enabled handheld breast ultrasound as a primary screening tool for breast cancer, particularly in low-resource settings. I offer the following comments to help strengthen the paper’s impact:

1. The introduction would benefit from a clearer framing of handheld ultrasound's role in breast cancer screening and why AI is needed. For instance, the authors can consider citing evidence that even without AI, handheld ultrasound can achieve high cancer detection sensitivity in low-resource settings. This underscores ultrasound's potential as an alternative to mammography in these regions.

2. The search strategy (PubMed and Google Scholar up to Dec 2023) is appropriate, but the authors should clarify if any other databases (e.g., EMBASE or IEEE Xplore) or gray literature were considered to capture relevant studies from both clinical and technical fields. Some AI studies appear only at conferences.

3. In the Methods, the manuscript groups studies by “AI task type, application time, and AI task.” For readers less familiar with AI jargon, the authors can define what is meant by frame selection, detection, segmentation, and classification tasks in the context of breast ultrasound.

4. The result that 79% of the studies were at high or unclear risk of bias is a crucial finding. The manuscript should elaborate on the main sources of bias observed. Were most studies lacking a proper independent validation cohort (index test bias), or were there concerns about patient selection (e.g., retrospective studies using case–control designs), or reference standard issues?

5. Following on the previous note, although the Discussion touches on the lack of robust evidence, it could more explicitly highlight the limitations of the studies reviewed. Key issues to discuss include: (a) the dearth of external validation (b) small or homogeneous datasets (c) spectrum and setting (many studies used datasets enriched with cancer cases or conducted retrospective analysis of clinical exams, which may overestimate performance compared to a true screening scenario with low prevalence); and (d) inconsistent reporting different studies use different performance metrics and thresholds (some report AUC, others report sensitivity at fixed specificity), which complicates direct comparison.

6. Nearly all the included studies seem to have evaluated AI algorithms on internal datasets or via cross-validation. This is a well-recognized shortcoming in AI diagnostics literature, as models often perform worse on truly independent data due to overfitting or population differences. The manuscript should stress that without external multi-center validation (ideally using prospective cohorts), the reported high performance of AI cannot be assumed to hold in new populations.

7. In line with generalizability, the authors should consider commenting on the diversity (or lack thereof) in the data used by the AI models. Did the review find that most models were trained on data from a single country or a single manufacturer’s equipment?

8. A critical aspect to address is how applicable the reviewed studies are to real-world screening scenarios (as opposed to diagnostic settings). The inclusion criteria captured studies on AI for BUS, but it’s not always clear if these studies simulated a screening context (meaning, asymptomatic women in population screening) or if they were focused on classifying known lesions in a clinical setting.

9. The Discussion should include a perspective on the clinical readiness of AI-informed handheld BUS tools. Despite many studies reporting high diagnostic accuracy, none of the AI systems reviewed appear to have been widely deployed in routine screening practice yet. It would be useful if the authors commented on what barriers remain before these AI tools can be used in the field. For instance, do any of the reviewed models have regulatory approval (like FDA clearance)? Are there ongoing prospective trials or implementation studies for AI-assisted ultrasound screening?

10. The authors have cited many pertinent studies; however, to strengthen the scholarly foundation, a few additional recent works could be referenced. For instance, Brot (Ultrasonography, 2024) provides a comprehensive review of AI in breast ultrasound and similarly conclude that more prospective studies are needed, and that AI could play a significant role in low-income regions.

11. Minor Points:

- Terminology: Use consistent terminology when referring to AI throughout the manuscript. Phrases like “AI-informed BUS” vs “AI-enhanced BUS” appear; it may be better to stick with one term (perhaps "AI-enhanced ultrasound") after initially defining it.

- The Conclusion section might be made slightly more cautious. Currently, the abstract concludes there has been “encouraging development,” but evidence “lacks robustness,” and that high-quality validation is key. As this work shows, while AI for handheld BUS shows promise and could be transformative for global breast cancer screening, substantial further research is required to confirm its accuracy and effectiveness in real-world screening programs. This tempered conclusion will ensure the stakeholders in global health do not take the current performance at face value.

Reviewer #3: This systematic review evaluates AI-enhanced handheld breast ultrasound (BUS) for cancer screening, particularly in low-resource settings. They do an excellent job classifying the tasks involved (framing, segmentation, interpretation), and find that while AI models show promising performance in tasks like lesion detection and classification, many studies lack robust validation on external datasets, limiting generalizability. The review emphasizes the need for high-quality validation, especially on diverse populations. Despite encouraging results, the authors conclude that further development and external validation are necessary before AI-enhanced BUS can be widely implemented in screening programs. Spot on! And the authors do it succinctly.

6. PLOS authors have the option to publish the peer review history of their article (what does this mean? ). If published, this will include your full peer review and any attached files.

**Do you want your identity to be public for this peer review?** For information about this choice, including consent withdrawal, please see our Privacy Policy .

Reviewer #1: No

Reviewer #2: No

Reviewer #3: **Yes: ** Felipe Batalini, M.D.

**Figure resubmission:**While revising your submission, please upload your figure files to the Preflight Analysis and Conversion Engine (PACE) digital diagnostic tool, https://pacev2.apexcovantage.com/. PACE helps ensure that figures meet PLOS requirements. To use PACE, you must first register as a user. Registration is free. Then, login and navigate to the UPLOAD tab, where you will find detailed instructions on how to use the tool. If you encounter any issues or have any questions when using PACE, please email PLOS at figures@plos.org. Please note that Supporting Information files do not need this step. If there are other versions of figure files still present in your submission file inventory at resubmission, please replace them with the PACE-processed versions. 
---

## [Decision Letter · Decision Letter 1]

2 Sep 2025

Artificial Intelligence-Enhanced Handheld Breast Ultrasound for Screening: A Systematic Review of Diagnostic Test Accuracy

PDIG-D-25-00126R1

Dear Dr. Bunnell,

We're pleased to inform you that your manuscript has been judged scientifically suitable for publication and will be formally accepted for publication once it meets all outstanding technical requirements.

Within one week, you'll receive an e-mail detailing the required amendments. When these have been addressed, you'll receive a formal acceptance letter and your manuscript will be scheduled for publication.

An invoice for payment will follow shortly after the formal acceptance. To ensure an efficient process, please log into Editorial Manager at https://www.editorialmanager.com/pdig/ click the 'Update My Information' link at the top of the page, and double check that your user information is up-to-date. For billing related questions, please contact billing support at https://plos.my.site.com/s/.

Kind regards,

Ismini Lourentzou

Section Editor

PLOS Digital Health

Additional Editor Comments (optional):

The authors have addressed most major concerns. Please consider incorporating the additional comments to strengthen the clarity of the paper, e.g., in terms of the selection of the literature resources, screening details, and stylistic/writing changes recommended by reviewers.

Reviewers' comments:

Reviewer's Responses to Questions

**Comments to the Author**

1. If the authors have adequately addressed your comments raised in a previous round of review and you feel that this manuscript is now acceptable for publication, you may indicate that here to bypass the “Comments to the Author” section, enter your conflict of interest statement in the “Confidential to Editor” section, and submit your "Accept" recommendation.

Reviewer #1: All comments have been addressed

Reviewer #2: All comments have been addressed

Reviewer #4: (No Response)

2. Does this manuscript meet PLOS Digital Health’s publication criteria ? Is the manuscript technically sound, and do the data support the conclusions? The manuscript must describe methodologically and ethically rigorous research with conclusions that are appropriately drawn based on the data presented.

Reviewer #1: Yes

Reviewer #2: Yes

Reviewer #4: Partly

3. Has the statistical analysis been performed appropriately and rigorously?

Reviewer #1: Yes

Reviewer #2: Yes

Reviewer #4: N/A

4. Have the authors made all data underlying the findings in their manuscript fully available (please refer to the Data Availability Statement at the start of the manuscript PDF file)?

Reviewer #1: Yes

Reviewer #2: Yes

Reviewer #4: Yes

5. Is the manuscript presented in an intelligible fashion and written in standard English?

PLOS Digital Health does not copyedit accepted manuscripts, so the language in submitted articles must be clear, correct, and unambiguous. Any typographical or grammatical errors should be corrected at revision, so please note any specific errors here.

Reviewer #1: Yes

Reviewer #2: Yes

Reviewer #4: No

6. Review Comments to the Author

Please use the space provided to explain your answers to the questions above. You may also include additional comments for the author, including concerns about dual publication, research ethics, or publication ethics. (Please upload your review as an attachment if it exceeds 20,000 characters)

Reviewer #1: (No Response)

Reviewer #2: I appreciate the time and effort that the authors have put into responding to the comments and incorporating the feedback into the revised manuscript.

I agree with the changes made and have no further comments.

Reviewer #4: Thank you for the opportunity to review this systematic review, which addresses the important topic of AI-enhanced breast ultrasound for breast cancer screening, a technology which holds potential in low-resource settings but, as the authors point out, currently has many limitations. The authors have revised the manuscript to address several important concerns raised by prior reviewers. However, some important issues remain that should be addressed before the manuscript can be considered for publication.

Introduction:

- The introduction (esp. line 69-70) of the manuscript frames malignancy detection by ultrasound as relying on successes or failures of radiologist perception/interpretation, which may be augmented by AI, but it seems to downplay the importance of image acquisition by a skilled sonographer in the context of US screening. It’s unclear in the framing of the article how AI might get around this. There is mention of frame selection (and only 1 study on this topic), but this seems to be, from the description, a step downstream of image acquisition. This issue should be addressed in the manuscript, including in the introduction and discussion. 

Methods: 

- The authors state they followed “much of the methods” from a prior review (Freeman et al.), but it do not specify which components (e.g., data extraction, bias assessment, synthesis strategy) were adopted versus modified. This should be explicitly clarified.

- The authors chose PubMed and Google Scholar as the sole databases but did not include EMBASE, Web of Science or Scopus, which are commonly included in systematic reviews in the biomedical sciences. Was any validation of the search method performed, e.g., whether known key studies were retrieved by this strategy? If not, this should be acknowledged as a limitation.

- It is unclear how the title/abstract screening was conducted. How many reviewers were involved? How were disagreements resolved? Were inclusion/exclusion criteria independently applied by multiple reviewers? This information should be clearly described.

- The term “unseen group of patients” is confusing. Does this refer to patients who were asymptomatic and intended to reflect a screening population? This phrase should be revised or clearly defined to avoid ambiguity.

General comments:

- Writing quality: There is notable tense confusion throughout the manuscript, particularly in the Methods section but also elsewhere, where present tense is often used for actions that were performed in the past (e.g. 96-105). Consistent use of past tense in describing study methods and results is expected in scientific writing. Careful line-by-line editing to correct this source of confusion is required to improve clarity of writing.

- Figure quality: The quality and resolution of several figures are insufficient and unreadable in the provided format. All figures should be re-uploaded in high resolution so that reviewers and future readers can adequately assess the presented data.

- English language limitation: The exclusion of non-English language studies should be mentioned in limitations. Though the authors cite references that suggest this should not affect results, it remains a limitation for research that intends to have a global health impact.

7. PLOS authors have the option to publish the peer review history of their article (what does this mean? ). If published, this will include your full peer review and any attached files.

**Do you want your identity to be public for this peer review?** For information about this choice, including consent withdrawal, please see our Privacy Policy . 

Reviewer #1: None

Reviewer #2: No

Reviewer #4: No
